# Simultaneous Determination for Nine Kinds of N-Nitrosamines Compounds in Groundwater by Ultra-High-Performance Liquid Chromatography Coupled with Triple Quadrupole Mass Spectrometry

**DOI:** 10.3390/ijerph192416680

**Published:** 2022-12-12

**Authors:** Shanshan Chen, Yi Zhang, Qinghua Zhao, Yaodi Liu, Yun Wang

**Affiliations:** 1School of Environment, Tsinghua University, No. 30 Shuangqing Road, Hai Dian District, Beijing 100084, China; 2SHANGHAI Soong Ching Ling School, Shanghai 200000, China; 3Physics, Tibet University, No. 10 Zangda East Road, Lhasa 850000, China; 4School of Water Resources and Environmental Engineering, Nanyang Normal University, No. 1398 Wolong Road, Nanyang 473061, China

**Keywords:** groundwater, solid phase extraction (SPE), ultra-high performance liquid chromatography triple quadrupole instrument (UHPLC-MS/MS), n-nitrosamine compounds

## Abstract

The ability to effectively detect N-nitrosamine compounds by liquid chromatography–tandem mass spectrometry presents a challenge due to the problems of high detection limits and difficulty in simultaneous N-nitrosamine compound detection. In order to overcome these limitations, this study reduced the detection limit of N-nitrosamine compounds by applying n-hexane pre-treatment to remove non-polar impurities before the conventional process of column extraction. In addition, ammonium acetate was used as the mobile phase to enhance the retention of nitrosamine target substances on the chromatographic column, with formic acid added to the mobile phase to improve the ionization level of N-nitrosodiphenylamine, to achieve the simultaneous detection of multiple N-nitrosamine compounds. Applying these modifications to the established detection method allowed the rapid and accurate detection of N-nitrosamine in water within 12 min. The linear relationship, detection limit, quantification limit and sample spiked recovery rate of nine types of nitrosamine compound were investigated, showing that the correlation coefficient ranged from 0.9985–0.9999, while the detection limits of the instrument and the method were 0.280–0.928 µg·L^−1^ and 1.12–3.71 ng·L^−1^, respectively. The spiked sample recovery rate ranged from 64.2–83.0%, with a standard deviation of 2.07–8.52%, meeting the requirements for trace analysis. The method was applied to the detection of N-nitrosamine compounds in nine groundwater samples in Wuhan, China, and showed that the concentrations of N-nitrosodimethylamine and NDEA were relatively high, highlighting the need to monitor water bodies with very low levels of pollutants and identify those requiring treatment.

## 1. Introduction

The safety of groundwater resources is important for public and environmental health, economic development and social stability [1,2]. The direct discharge or ineffective treatment of chemical sewage and wastewater, as well as the excessive agricultural use of nitrogen containing pesticides, fertilizers and other chemical products, has resulted in the migration of N-nitrosamine compounds and their precursors into groundwater resources through the soil, causing a serious deterioration in groundwater quality [3,4,5].

N-Nitrosamine compounds are highly toxic and carcinogenic, having been listed by the International Research Agency on Cancer as one of the three most serious potential human carcinogens [6]. In particular, the American Toxic Substances and Disease Registry of the U.S. Department of Health and Human Services found that N-nitrosodiphenylamine (NDPhA) can cause bladder cancer in rats [7]. Furthermore, the U.S. Environmental Protection Agency (US EPA) reported that even at a trace level of 0.7 ng·L^−1^, N-nitrosodimethylamine (NDMA) exposure may lead to a risk of cancer [8,9]. The detection of N-nitrosamine compounds has become a mandatory aspect of the water treatment process in many countries. For example, it has been widely performed and studied in drinking water supplies in Canada, the US and Japan [10,11]. However, in view of the relatively low concentration of N-nitrosamines observed in polluted groundwater worldwide and the unique difficulties presented by groundwater pollution detection, the monitoring of N-nitrosamines in groundwater resources poses a major challenge [12,13,14,15,16].

N-Nitrosamines (NAs) are small-molecule and highly polar compounds containing nitrogen. The physicochemical properties of common N-nitrosamines are shown in Table 1 [16]. The physical and chemical properties of N-nitrosamines are vary greatly; most can dissolve in organic solvents such as alcohol and dichloromethane, and only some compounds can dissolve in water. Therefore, the quantification of N-nitrosamines is very difficult [17,18,19].

Low-concentration N-nitrosamine detection is currently a key focus in environmental research [20,21]. GC–MS is the most commonly applied method for the detection of N-nitrosamine compounds [22,23,24,25]. For example, using the 521 method of the US EPA, GC–MS is capable of detecting six types of N-nitrosamine compound, such as NDMA, requiring a minimum concentration in drinking water of 1–2 ng·L^−1^ to be achieved using SPE column pre-concentration and enrichment [26]. Mousa Amayreh (2019) developed an automated headspace solid-phase microextraction coupled with gas chromatography–mass spectrometry (automated HS-SPME/GC–MS) for the determination of four N-nitrosamines—N-nitrosodiethylamine N-nitrosodi-n-propylamine, N-nitrosopiperidine and N-nitrosodi-n-butylamide—in groundwater samples and determined the N-nitrosamine in groundwater samples from different locations in Saudi Arabia [27]. However, this method could not detect thermally unstable nitrosamine compounds such as NAPhA [28,29]. Liquid chromatography can be used to analyze and detect N-nitrosamine compounds that easily decompose and are difficult to volatilize into a gas when heated, allowing the determination of substances that cannot be detected by gas chromatography [30]. Liquid chromatography tandem mass spectrometry (LC–MS) is a highly accurate method for the determination of trace compounds in complex sample matrices, overcoming the problem of the insufficient sensitivity of liquid chromatography for the detection of small-molecule nitrosamines [31,32]. Cristina Ripollés et al. (2017) used LC–MS to determine eight types of N-nitrosamine substances in water (not NDPhA), achieving detection limits of 1–8 ng·L^−1^; NDPhA was not detectable using this method [33]. In contrast, Kadmi et al. (2013) were only able to determine four types of nitrosamine compounds (NDPA, NMOR, NMEA and NDMA) in water using LC–MS/MS [34]. Ji-Hyun et al. (2019) used LC–MS/MS (APCI) to determine nine types of N-nitrosamine in water, achieving recovery rates of >70%, except for NDPhA which had a recovery rate of <50% [35]. Arnaud Djintchui Ngongang (2015) developed a methodology for the analysis of nine N-nitrosamines based on ultra-high-performance liquid chromatography (UHPLC) coupled to mass spectrometry using heated electrospray ionization (HESI) in positive ionization mode with a Q-Exactive mass spectrometer. The extraction recoveries in real matrices ranged from 68–83% for eight of the nine target nitrosamines and values of 22–31% for NDPhA; the detection limits ranged from 0.4 to 12 ng·L^−1^ [36]. Among the currently available LC–MS detection methods, few approaches allow the simultaneous determination of nine types of nitrosamine compound and the detection limits are typically relatively high. Furthermore, few studies have described NDPhA detection methods. The existing methods for NDPhA detection are limited by problems such as low recovery rates. Although the conventional elution process can effectively remove polar impurities using activated carbon extraction columns, it does not effectively remove non-polar impurities. As a result, non-polar impurities remain in the eluted concentrated solution along with the target substances, resulting in interference from the sample matrix and increasing the detection limit of nitrosamine substances [37,38,39]. Therefore, the detection limit for nitrosamine substances can be effectively reduced by improving the removal of non-polar impurities during the conventional column extraction process [40]. 

Volatile small-molecule nitrosamine substances, such as NDMA, become extremely volatile when methanol is used as the solvent, making it impossible to maintain effective analysis and testing over the long-term. Pure water is commonly used as the solvent during the extraction of nitrosamines from water [41]. However, the insolubility and volatility of NDPhA in water results in a relatively poor detection response using APCI sources, leading to a low recovery rate for NDPhA and difficulty in simultaneously detecting other nitrosamine substances [42]. Therefore, it is necessary to find a suitable solvent that allows NDPhA solubility and reduces the volatility of other nitrosamine substances to realize the simultaneous detection of nine types of nitrosamine compound.

In view of the aforementioned problems, this study modified the conventional analysis process, with the aim of removing non-polar impurities from samples prior to the conventional column extraction process [43]. In addition, ammonium acetate was used as the mobile phase to enhance the retention of nitrosamine target substances by the chromatographic column, with formic acid added to the mobile phase to improve the ionization level of NDPhA [44]. The pH value of the mobile phase was controlled with the aim of ensuring that target substances existed in an ionic form, reducing peak deformation and splitting caused by the coexistence of ionic and molecular forms, inhibiting silanol group activity, and preventing the tailing of alkaloids in ionic form, while ensuring that the retention time of the target substances was not affected by acidity [45,46]. Finally, the intensity and peak pattern of the target substance mass spectrum response were investigated with varying formic acid concentrations in the mobile phase.

## 2. Materials and Methods

### 2.1. Reagents and Chemicals

A methanol solution (2000 mg·L^−1^ each component) containing N-nitrosodimethylamine (NDMA), N-nitrosomethylethylamine (NMEA), N-nitrosodiethylamine (NDEA), N-nitrosodi-n-propylamine (NDPA), N-nitromorpholine (NMor), N-nitrosopyrrolidine (NPyr), N-nitrosopiperidine (NPip), N-nitrosodi-n-butylamide (NDBA), N-nitrosodiphenylamine (NDPhA) were purchased from Sigma–Aldrich (Milan, Italy). Ammonium acetate was LC–MS grade from Merck (Darmstadt, Germany), and methanol, n-hexane dichloromethane and formic acid were LC–MS grade from Fisher (Pittsburgh, Pennsylvania, PA, USA). All ultra-pure water used in the experiment was prepared by a Milli-Q system (Millipore, MA, USA). 

### 2.2. UHPLC-MS/MS Conditions

The UHPLC–MS/MS system consisted of automatic sampling system (Shimadzu SIL-30AC), ultra-high performance liquid chromatography (Shimadzu LC-30AD), a triple quadrupole tandem mass spectrometer (Shimadzu LCMS8060) and ESI source (Shimadzu Japan). For quantitative analysis, a Shim-pack GIST C18 chromatographic column (2 µm, 2.1 mm I.D. × 100 mm L) was used. The mobile phase composed of pure methanol (A), and 5 mmol·L^−1^ ammonium acetate aqueous solution (containing 0.1% formic acid) (B). Liquid chromatography operating parameters and mobile phase gradient conditions are shown in Table 2. The flow rate was set at 0.3 mL·min^−1^. The column temperature was set at 40 °C and the injection volume was 10 µL. The column temperature was set at 30 °C, and the sample tray temperature was maintained at 15 °C.

All components were directly determined by MS/MS after liquid chromatography separation. For MS/MS detection, high-purity nitrogen was used as the nebulizer and the drying gas. The nebulizer gas was set at 3.0 mL·min^−1^ and the drying gas was set at 12.0 mL·min^−1^. Air was used as the heating gas and set 8.0 mL·min^−1^. Argon was used as the collision activation dissociation (CAD) gas. The ESI source was operated under the positive ion ESI (+) multiple selective reaction monitoring conversion (MRM) mode. The following parameters were used: interface temperature, 250 °C; DL temperature, 290 °C; heating module temperature, 350 °C; delay time, 3.0 ms; dwell time, 12.0 ms; and MRM parameters are shown in Table 3. Data acquisition was performed using MRM mode, monitoring two pairs of MRM conversion ion pairs. The instrument was controlled by Labsolution 5.93 (Shimadzu, Kyoto, Japan). Under the same analytical conditions as the calibration standard sample, the qualitative confirmation of the target substance in the sample was mainly based on two factors: (1) relative to the calibration standard, a retention time difference within 0.2 min; (2) the ratio difference between two pairs of MRM conversions of the target substance relative to the calibration standard sample was within 20%. For the quantification of the target substance, each nitrosamine substance was quantified by using MRM-converted ion pair with high abundance or less background interference, and the calibration curve was performed using the external standard peak area method [47].

### 2.3. Sample Extraction

Water samples (500 mL) were filtered by glass fiber filter paper (0.7 µm) to avoid blocking the solid phase extraction column. All the target substances were extracted by a solid phase extraction column (Resprep EPA Method 521, 2 g/6 mL, Milford, MA, USA) through an automatic solid phase extraction instrument. In order to enable the analyte to be in close contact with the solid surface, facilitate adsorption, and at the same time remove impurities in the column and reduce pollution, the solid phase extraction column was wetted and activated in turn with 10 mL n-hexane, 20 mL dichloromethane and 20 mL methanol. Then, the solid phase extraction column was washed with 20 mL ultra-pure water to make the sample solution in good contact with the adsorption surface and improve the extraction efficiency [48]. The water sample passed through the activated solid phase extraction column at a flow rate of 15 mL·min^−1^. Then, the solid phase extraction column was washed with 10 mL ultra-pure water and dried with high-purity nitrogen to remove the water in the column. The solid phase extraction column was eluted with 15 mL dichloromethane and dried; the eluent was concentrated to near dryness by nitrogen blowing (keeping the liquid level slightly fluctuating) and diluted to 0.5 mL with ultra-pure water (containing 25% methanol). The samples were filtered by 0.2 µm filter membrane to remove particulate matter before detection, and then analyzed by UHPLC-MS/MS. 

### 2.4. Collection and Preservation of Water Samples

Nine groundwater samples were selected from different places in Wuhan, China, and named Sample 1, Sample 2, Sample 3, Sample 4, Sample 5, Sample 6, Sample 7, Sample 8 and Sample 9. On-site sampling of each groundwater sample was carried out, and all the water samples were placed in brown glass bottles (500 mL) avoiding headspace and sealed with Teflon lined caps after addition of 50 mg of Na_2_S_2_O_3_ (Sigma–Aldrich, Italy) for dechlorination [49,50]. The samples were stored at 4 °C and analyzed within 4 days [51].

## 3. Results and Discussion

### 3.1. Optimization of Chromatographic Conditions

The signal response of the target compound on the ESI source may depend largely on the liquid chromatographic conditions, so the mobile phase proportions and column models were investigated. In the experiment, the mobile phase consisted of methanol and 5 mmol·L^−1^ ammonium acetate aqueous solution. To obtain a better target response value, various concentrations (0, 0.1%, 0.2%) of formic acid were added to the aqueous ammonium acetate solution, and the standard solution was determined at a mobile phase flow rate of 0.3 mL·min^−1^. The results showed that when 0.1% formic acid was added to the aqueous phase, the mass spectrometry response was the highest, and the peak type was the best. Therefore, methanol and 5 mmol·L^−1^ ammonium acetate aqueous solution (containing 0.1% formic acid) were used in the mobile phase.

In order to obtain the maximum sensitivity and optimize the chromatographic peak shape and resolution, ACQUITY UPLC C18 (2.1 mm × 50 mm, 1.8 µm) and Shim-pack GIST C18 (2.1 mm × 100 mm, 2 µm) were selected in this experiment. It was found that the two chromatographic columns had good separation and peak shape, but the ACQUITY UPLC C18 (2.1 mm × 50 mm, 1.8 µm) chromatographic column was easily blocked and the liquid phase pressure was too high. Therefore, the Shim-pack GIST C18 (2.1 mm I.D. × 100 mm L, 2 µm) chromatographic column was selected because it can significantly increase the sample flux. It can be seen from Figure 1 that all the target analytes have good chromatographic peaks and could be eluted from the chromatographic column within 12 min; the detection sensitivity and efficiency of the target compounds met the research needs. The determined chromatographic conditions were as follows: a mobile phase composed of pure methanol (A) and 5 mmol·L^−1^ ammonium acetate aqueous solution (containing 0.1% formic acid) (B). Liquid chromatography operating parameters and mobile phase gradient conditions are shown in Table 2. The flow rate was set at 0.3 mL·min^−1^. The column temperature was set at 40 °C and the injection volume was 10 μL. The column temperature was set at 30 °C and the sample tray temperature was maintained at 15 °C.

### 3.2. Optimization of Mass Spectrometry Conditions

To simultaneously and rapidly detect the nine N-nitrosamines in drinking water, we optimized the multiple reaction monitoring (MRM) method for each N-nitrosamine for greater sensitivity and selectivity. The pretest showed that the ESI (+) mode could get a better signal response than the ESI (−) mode, so the ESI (+) mode was used in this study. Using 50 µg·L^−1^ standard solution, the mass spectrometry conditions were further optimized by automatic optimization to obtain the parent ion, daughter ion, optimal cone voltage and collision energy of the target compound. The results are shown in Table 3. When analyzing the target substance, the protonated molecular ion peak ([M+H]^+^) was selected as the parent ion because of its high abundance. The N=O group and the =O group were removed during the protonation of the target substance. The two fragment ions of these compounds were [M-N=O-H]^−^ and [M^−^=O-H]^−^, respectively; they were selected as daughter ions [52].

### 3.3. Validation of the Quantitative Method

The validation process was performed using the criteria from the International Conferences of Harmonization (ICH), more specifically the Q2 (R1) guidelines [53]. The validation was performed to evaluate the NA analytical method in terms of the following parameters: linearity, precision, accuracy (% bias), instrumental detection limit, method detection limits and quantification limits. The recovery of the extraction procedure was also calculated for the nine target nitrosamines [54].

Nine N-nitrosamine calibration mixed solutions were prepared with 25% methanol aqueous solution, with the concentrations of each substance were set to 1.0, 5.0, 10.0, 50.0 and 100.0 µg·L^−1^. The calibration solutions of N-nitrosamine compounds were determined according to the analysis conditions in Section 2.2. The calibration curve was made by an external standard method with concentration as the abscissa and peak area as the ordinate (Table 4). Table 4 shows that the correlation coefficients of the nine N-nitrosamine compounds range from 0.9985 to 0.9999, and the linear correlation coefficients are good. The instrument detection limits and method detection limits (MDL) of the nine N-nitrosamine compounds are 0.280–0.928 µg·L^−1^ and 1.12–3.71 ng·L^−1^, respectively. 

The groundwater samples were used for standard addition experiments. The groundwater and matrix spiked sample solutions were obtained according to the preparation method in Section 2.3. The sample spiked content was 20 µg·L^−1^, and the parallel determination was performed six times. The recovery rate and standard deviation results are shown in Table 5. The recovery rate of the nine N-nitrosamine compounds was 64.2~83.0%, with a standard deviation of 2.07~8.52%, which meets the requirements for trace analysis. Moreover, the higher the concentration of the standard addition, the higher the recovery rate. 

### 3.4. Analysis of Actual Water Samples

According to the analysis method established above, nine groundwater samples were sampled and detected in Wuhan, China, and the distribution map of N-nitrosamine in the groundwater of the city was obtained (Figure 2). At least one kind of N-nitrosamine was detected within all nine groundwater samples. However, low concentrations of N-nitrosamines were observed in the groundwater samples. From Figure 2, it can be seen that six N-nitrosamine were mainly detected in the nine samples. It is worth noticing that six types, including NDMA, NPYR, NDEA, NDPA, NDBA and NDphA, were detected in the groundwater samples with concentrations ranging from 4.81~60.99 ng·L^−1^, 2.29~12.14 ng·L^−1^, 1.86~40.60 ng·L^−1^, 0~1.47 ng·L^−1^, 0~8.22 ng·L^−1^, 0~1.13 ng·L^−1^. NDMA, NDEA and NPYP were all detected in all nine samples; the concentration of NDMA was the highest. It could be concluded that the kinds of nitrosamine disinfection by-products in the groundwater of this city are more concentrated, but the detection rate is high and the content is generally low. The identified types have also been previously observed in the Wuhan section of the Yangtze River, suggesting that various kinds of N-nitrosamines may pollute groundwater through surface water [51,54,55,56]. The average concentrations of the nine N-nitrosamine detected are less than the World Health Organization’s drinking water limit of 100 ng·L^−1^, but NDMA and NEMA greatly exceed the corresponding concentrations of 0.7 ng·L^−1^ and 0.2 ng·L^−1^ with carcinogenic risks [8,9,10,11,12]. The concentrations of the other seven nitrosamines are lower than the corresponding concentrations of carcinogenic risk levels in drinking water.

## 4. Conclusions

In this study, nine types of N-nitrosamine compound (including unstable NDPhA) were investigated in groundwater environmental water samples, and simultaneous detection using triple quadrupole mass spectrometry and ultra-high performance liquid chromatography was used. The method was optimized effectively, by adjusting the pretreatment extraction column packing material and mobile phase components. 

The detection method established in this study was used to determine nine types of N-nitrosamine compounds in calibrated solutions. Within the linear range, all correlation coefficients were greater than 0.998, exhibiting good linearity. The instrument detection limit and method detection limit were 0.280–0.928 µg·L^−1^ and 1.12–3.71 ng·L^−1^, respectively, effectively meeting the requirements of the World Health Organization (WHO) and various relevant national standards. The results of spiked recovery tests using groundwater samples exhibited recovery rates from 64.2–83.0%, with standard deviations of 2.07–8.52%, meeting the requirements for trace analysis.

The analysis of nine groundwater samples using the optimized method showed that many kinds of nitrosamine disinfection by-products are present in groundwater in the study region, although generally the content was relatively low, with average concentrations all below the drinking water limit of 100 ng·L^−1^ specified by the WHO. However, the NDMA and NEMA concentrations were relatively high, exceeding the concentrations of 0.7 ng·L^−1^ and 0.2 ng·L^−1^ and corresponding to an increased cancer risk [8,9,10,11,12]. 

Therefore, treatment techniques for minimization of N-nitrosamines need to be developed. Further attention needs to be paid to the monitoring and treatment of groundwater with very low levels of pollutants.

## Figures and Tables

**Figure 1 ijerph-19-16680-f001:**
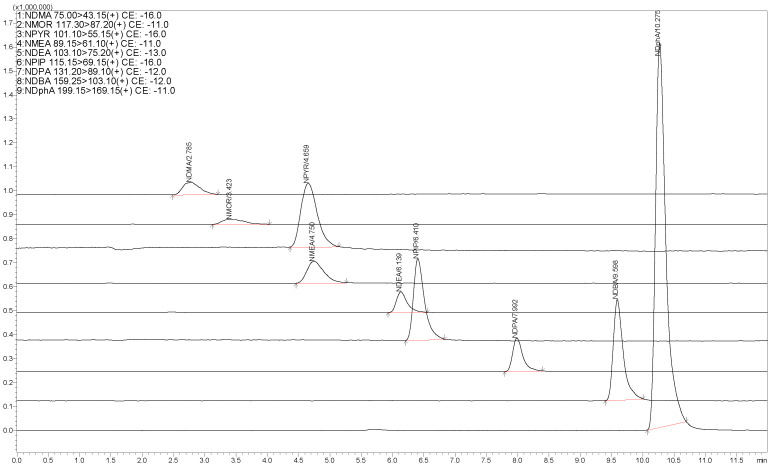
MRM chromatogram of N-nitrosamine compounds (50.0 µg·L^−1^).

**Figure 2 ijerph-19-16680-f002:**
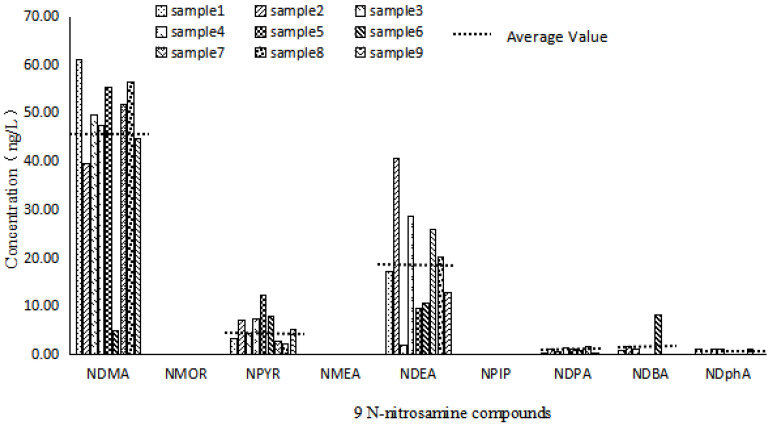
Distribution of 9 nitrosamines in 9 samples.

**Table 1 ijerph-19-16680-t001:** Physicochemical properties of common N-nitrosamines.

Compound	Abbr.	Molecular Formula	Structural Formula	Boiling Temperature (°C)	Carcinogenicity
N-nitrosodimethylamine	NDMA	C_2_H_6_N_2_O	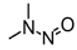	153	2A ^a^
N-nitromorpholine	NMOR	C_5_H_10_N_2_O	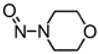	226.1	2B ^b^
N-nitrosopyrrolidine	NPYR	C_4_H_8_N_2_O	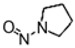	214	2B
N-nitrosomethylethylamine	NMEA	C_3_H_8_N_2_O	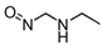	154.4	2B
N-nitrosodiethylamine	NDEA	C_4_H_10_N_2_O	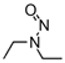	173.9	2A
N-nitrosopiperidine	NPIP	C_5_H_10_N_2_O	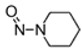	229.8	2B
N-nitrosodi-n-propylamine	NDPA	C_6_H_14_N_2_O	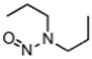	206	2B
N-nitrosodi-n-butylamide	NDBA	C_8_H_18_N_2_O	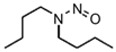	250	2B
N-nitrosodiphenylamine	NDPhA	C_12_H_10_N_2_O	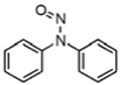	268	—

^a^ There is limited evidence of carcinogenicity in humans. ^b^ Animal evidence is sufficient but human data is insufficient.

**Table 2 ijerph-19-16680-t002:** Liquid chromatography operating parameters and mobile phase gradient conditions.

Time (min)	Module	Command	Value
0.01	Pumps	Pump B Conc.	90
2.50	Pumps	Pump B Conc.	40
10.00	Pumps	Pump B Conc.	40
10.01	Pumps	Pump B Conc.	90
12.00	Pumps	Stop	

**Table 3 ijerph-19-16680-t003:** Mass spectrometric parameters of N-nitrosamines.

Compound	CAS No.	Interface Voltage (KV)	Precursor Ions	Product Ions	Q1 Pre Bias (V)	CE (V)	Q3 Pre Bias (V)
NDMA	62-75-9	1.0	75.15	58.10 *	−13	−15	−22
				43.10	−13	−16	−16
NMOR	59-89-2	4.5	117.30	87.20 *	−13	−11	−13
				45.00	−19	−18	−17
NPYR	930-55-2	4.5	101.10	55.15 *	−17	−16	−20
				39.05	−17	−30	−13
NMEA	624-78-2	4.5	89.15	61.10 *	−17	−11	−23
				43.10	−15	−18	−16
NDEA	55-18-5	4.5	103.10	75.20 *	−17	−13	−30
				47.20	−20	−15	−19
NPIP	100-75-4	4.5	115.15	69.15 *	−20	−16	−28
				41.05	−20	−23	−14
NDPA	621-64-7	4.5	131.20	89.10 *	−22	−12	−16
				43.10	−24	−14	−16
NDBA	924-16-3	0.5	159.25	103.10 *	−11	−12	−24
				57.15	−11	−14	−16
NDPhA	86-30-6	0.5	199.15	169.15 *	−13	−11	−10
				66.10	−13	−24	−10

* Represents quantitative ion pair.

**Table 4 ijerph-19-16680-t004:** Calibration curve parameters of nine N-nitrosamine compounds.

Compound	Regression Equation	R^2^	LOD (µg·L^−1^)	MDL (ng·L^−1^)
NDMA	Y = 9577.26X + 5759.90	0.9999	0.928	3.71
NMOR	Y = 5047.71X − 13,728.0	0.9992	0.546	2.19
NPYR	Y = 47,890.0X − 43,846.3	0.9985	0.639	2.56
NMEA	Y = 16,819.2X − 15,053.7	0.9996	0.280	1.12
NDEA	Y = 10,235.0X − 3454.58	0.9990	0.513	2.05
NPIP	Y = 41,240.2X − 12,518.2	0.9995	0.717	2.87
NDPA	Y = 15,371.4X − 2776.90	0.9998	0.481	1.92
NDBA	Y = 45,940.36X − 14,909.9	0.9997	0.766	3.07
NDPhA	Y = 184,657X − 168,651	0.9994	0.912	3.65

**Table 5 ijerph-19-16680-t005:** Results of recovery rate (n = 6).

Compound	Recovery Rate (%)	RSD (%)
NDMA	71.4	2.65
NMOR	69.2	3.48
NPYR	65.1	2.07
NMEA	70.6	8.52
NDEA	64.2	3.92
NPIP	70.6	2.96
NDPA	68.0	2.43
NDBA	67.2	4.85
NDphA	66.0	6.41

## Data Availability

The data that support the findings of this study are available from the corresponding author, Wang Yun reasonable request.

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
