# Peer review of "Simultaneous Determination for Nine Kinds of N-Nitrosamines Compounds in Groundwater by Ultra-High-Performance Liquid Chromatography Coupled with Triple Quadrupole Mass Spectrometry"

_ijerph, 2022, doi:10.3390/ijerph192416680_

Round 1

Reviewer 1 Report

Comments:

1. The reason for each step of 2.3. Sample extraction please be informed clearly.

2. The batch or lot No. of each 9 nitrosamines should be addressed.

3.  [Error! Reference source not 183 found.]. is evident at line 182-183.

4. Is there any statistical analysis technique to indicate in part of Methodology?

5. Literature review more with some discussion or term of comparison in Introduction for this topic should be conducted relatively such as "Analysis of nine N-nitrosamines using liquid chromatography-accurate mass high resolution-mass spectrometry on a Q-Exactive instrument" Analytical Methods 7:5748-5759; Determination of N-nitrosamines in Water by Automated Headspace Solid-Phase Microextraction

  • 2018
  • DOI: 
  • 10.1007/s13369-018-3567-6; 

Application of chromatographic techniques in the analysis of total nitrosamines in water, https://doi.org/10.1016/j.heliyon.2020.e03447 and so on.

6. Why only 9 water samples of ground waters were used and how is differences for them in term of location nearby or environmental concern.? Owing to its used as drinking water, so how are they treated before selling or distribution.  Are they appropriate sample as representation any point of view as good samples and enough for any claimation?

7. In term of method validation of analysis, how your experimental design cover for all criteria? If have not yet please include for confident of analysis.

8. The specific name of city for southern city of China should be addressed if the authors think that they all are good representatives or why have to be these cities.

9. S.D. should be included in bar of Figure 2. 

10. The more discussion with supporting references should be conducted for 3.4. Analysis of actual water samples.

11. Chemical structure of nine compounds should be included such as in Introduction and employed for discussion to clarify and related to the analysis result based on their properties relied on their basic structure.

12. Concise conclusion as single paragraph should be stated with the suggestion of treatment techniques for minimization of nitrosamine contaminants.

13. Current published related references should be included and employed in part of Introduction and discussion.

Reviewer 2 Report

This manuscript describes the detection limit of N-nitrosamine compounds using n-hexane pre-treatment to remove non-polar impurities before the conventional process of column extraction. In addition, ammonium acetate was used as the mobile phase to enhance the retention of nitrosamine target substances on the chromatographic column, with formic acid added to the mobile phase to improve the ionization level of N-nitrosodiphenylamine, achieving the simultaneous detection of multiple N-nitrosamine compounds.

Major comments:

1. Where the sample was collected and what is its specific information? How can authors ensure that the samples collected are representative?

2. In order to increase the relevance of this study to the journal, it is recommended that the authors increase the number and scope of experimental samples in this study to further increase the depth of the study, and if other methods can be used to further verify the conclusions of this study, it will make this study more interesting.

Minor comments:

1. Your manuscript on the advantages and novelties of the detection of n-nitrosamine compounds to the published literature, such as the time required for the test and the conditions necessary for the test, should be described in more detail in the abstract, introduction, and discussion sections.

2. Table 1 in the manuscript is not standardized, the number of significant digits reserved in the description of time is not uniform and needs to be corrected.

3. The reference to line 182 of the manuscript is unfounded, please pay attention to the rigor and standardization of the writing.

Round 2

Reviewer 1 Report

accept

Reviewer 2 Report

No more comments.